# Satisfaction Level and Performance of Physiotherapy Students in the Knowledge of Musculoskeletal Disorders through Nearpod: Preliminary Reports

**DOI:** 10.3390/ijerph20010099

**Published:** 2022-12-21

**Authors:** Maria Jesus Vinolo-Gil, Ismael García-Campanario, Carolina Lagares-Franco, Gloria Gonzalez-Medina, Manuel Rodríguez-Huguet, Francisco Javier Martín-Vega

**Affiliations:** 1Department of Nursing and Physiotherapy, Faculty of Nursing and Physiotherapy, University of Cadiz, 11009 Cadiz, Spain; 2Research Unit, Biomedical Research and Innovation Institute of Cadiz (INiBICA), Puerta del Mar University Hospital, University of Cadiz, 11009 Cadiz, Spain; 3Rehabilitation Clinical Management Unit, Interlevels-Intercenters Hospital Puerta del Mar, Hospital Puerto Real, Cadiz Bay-La Janda Health District, 11006 Cadiz, Spain; 4Group PAIDI UCA CTS391, Faculty of Medicine, University of Cadiz, 11003 Cadiz, Spain; 5INiBICA Group CO15 Population and Health: Determinants and Interventions, PAIDI UCA Group: CTS553, Department of Statistics and Operations Research, Faculty of Medicine, University of Cadiz, 11510 Cádiz, Spain; 6CTS-986 Physical Therapy and Health (FISA), University Institute of Research in Social Sustainable Development (INDESS), 11009 Cadiz, Spain

**Keywords:** physiotherapy, musculoskeletal disorders, Nearpod, problem-based learning, education

## Abstract

Physiotherapists are at high risk for musculoskeletal disorders. There is a need in academia to address workers’ health issues at the time of graduation. Nearpod is an educational application founded on a web-based learning tool. In the field of Health Sciences, the use of Nearpod has been scarce. The objective of this study was to determine the level of satisfaction with using this interactive tool and to assess the influence of using Nearpod in class on students’ performance while dealing with the topic of musculoskeletal disorders in third-year Degree in Physiotherapy students during the 2021–2022 academic year. The participants were students at the University of Cadiz. They were randomly divided into two groups, a control group using a PowerPoint presentation and an experimental group using the interactive Nearpod application. The experimental group took two surveys to determine their satisfaction with the method used. Students also took a multiple-choice test to assess the knowledge acquired. In the surveys, a high percentage of satisfaction was obtained (97.62% and 99.39%). There were no significant differences in the scores obtained by the two groups, although there were significant differences in response time in favor of the experimental group (*p* = 0.022). Although studies should be carried out with larger samples and on different subjects, it seems that Nearpod is a tool with great potential for teaching the study of musculoskeletal disorders.

## 1. Introduction

Musculoskeletal disorders (MSDs) are injuries (physical and functional alterations) associated with the locomotor system: muscles, tendons, ligaments, nerves, or joints [1]. They are important occupational health problems and cause high economic costs [2]. MSDs are multifactorial and may develop due to continuous and prolonged exposure of workers to noxious and harmful effects in the work environment [3]. Their symptoms may occur alone or concomitantly, with the main ones being pain and discomfort, mainly in the neck, shoulders, cervical and lumbar regions, and lower limbs [4]. Physically demanding tasks are among the risk factors most associated with MSDs, particularly, awkward postures, handling of loads, application of forces, and repetitiveness of actions. A report carried out in 2014 by the National Institute for Safety and Health at Work in Spain commented that healthcare workers were some of the most frequently affected because many of these actions are performed on a daily basis [5]. Therefore, physiotherapists are at high risk for MSDs [6].

In academia, several studies have been carried out on Health Science students, highlighting the need to address issues related to workers’ health at graduation [7,8,9]. Back pain has been found in the areas of health in university populations. Factors related to academic activity and those derived from professional activities, in addition to sociodemographic and psychosocial factors, interact and condition its manifestation [10].

In addition, the prevalence of these symptoms among students points to the need to implement preventive and health promotion actions to contribute to a better quality of life and health, both academically and in the future, as professionals. For this reason, it would be advisable to learn about musculoskeletal disorders during the Physiotherapy degree [11].

The pandemic caused by the current coronavirus, COVID-19, has increased the use of new technologies [12], and students can access online learning tools and interactive resources [13] ubiquitously with their own devices, such as laptops, smartphones, or tablets, for lifelong studying [14,15]. Hence, technology-enhanced learning has become a common feature of higher education [16], with 80% of university students using a smartphone or a tablet to study [17].

Regarding what these tools should look like, in a study carried out to find out the opinion of university students on different models of presentation with Information and Communication Technologies (ICTs) tools, it was found that their design has to be more visual, with interactive content, interspersing tasks of verification and consolidation of knowledge [18].

Online tools such as Kahoot (Kahoot!, Trondheim, Norway), Socrative (Showbie Inc., Edmonton, AB, Canada), Quizlet Live (Quizlet, San Francisco, CA, USA), and Nearpod (Dania Beach, FL, USA), among others, have been used by teachers in their classrooms to develop cognitive, motivational, emotional, and social skills [19]. Specifically, the use of interactive tools by undergraduate physiotherapy students has had positive results for the development of clinical reasoning skills [20], improved participation, and reduced absenteeism [21], and it is useful for studying manual therapy [22]. In a recent meta-analysis of digital learning designs within physiotherapy education, significant differences in favor of practical skills were found with interactive websites [23].

Nearpod is an educational application based on a web-based learning tool. It is a cloud-based educational application that can be used in conjunction with video conferencing tools to effectively engage students in a synchronous online classroom [24]. Students become active learners, improving their involvement [25,26]. It also allows teachers to evaluate students’ responses and subsequently draw meaningful conclusions, which can help them plan future lessons and guide further teaching [27].

This teaching–learning tool has been used and analyzed in other areas, such as Computer Science [14], Information Science [28], Financial Accounting [29], and Pharmacy and Bioscience [30]. In the field of Health Sciences, the use of Nearpod has been rare; however, it has been used in Nursing to determine whether the frequency of participation with this pedagogical tool was associated with students’ grades, with significant results [31]. However, the degree of satisfaction has never been analyzed. To our knowledge, no articles have analyzed this type of tool for the study of musculoskeletal disorders in Physiotherapy Degree students.

Based on this, the objective of this study was to determine the level of satisfaction of Physiotherapy Degree students with their use of this interactive tool and to assess the influence of using Nearpod in class on students’ performance for teaching in the study of musculoskeletal disorders.

## 2. Materials and Methods

This is an intervention study carried out in a group of Health Science students. The sample consisted of 55 subjects in the third year of the Physiotherapy Degree at the University of Cadiz. The research was carried out during the first semester of the 2021–2022 academic year in the classes of the subject “Physiotherapy in Clinical Specialties II”. The subject deals with physiotherapy in the main musculoskeletal disorders.

### 2.1. Procedure

The subjects were randomly divided into two groups by means of opaque envelopes. These were given to a professor who was not involved in the project. He was in charge of shuffling the envelopes and handing them out randomly to the students. The envelopes were marked with a “C” or “N.” Students with a “C” were assigned to the control group (*n* = 27), and those with an “N” on the envelope were assigned to the experimental group (*n* = 28). On the day of the test, 5 students from the control group were absent from class, leaving 28 students in the group that had the Nearpod presentation and 22 in the group that had the PowerPoint presentation (Figure 1). This project was carried out during two hours of theory. In one of the hours, the control group was with a subject teacher conducting a discussion on a topic unrelated to musculoskeletal disorders, while one of the project teachers was in another classroom with the experimental group using Nearpod. In the next hour, the control group went on to teach the musculoskeletal disorders class in the traditional way with the project teacher, and the experimental group went to the second classroom to conduct the discussion with the other subject teacher.

For the implementation of the project, Nearpod was used in a theoretical class in the experimental group. This teaching tool allows real-time quizzes, slides, videos, drawings, surveys, open-ended questions, web content, and various activities [25,26] to be presented through an interactive presentation created and controlled by teachers [14]. It can also be used as a formative assessment tool, as it is possible to obtain immediate feedback from students, or as an interactive tool, as students can participate in the content of the presentations through their answers, comments, or questions presented [32].

The students received the contents of the subject, specifically, “Musculoskeletal disorders in physiotherapists”, sending their answers using the application, and the teacher supervised the activities, receiving the results from each student in real-time.

The learning method was active and focused on providing the content of the subject in a way that increased participation and interactivity with the students. They were recruited through the virtual campus when they were given the timetable for the theory class, and it was explained to them that they should access the Nearpod platform. They had to access “nearpod.com” and, in the “Students, Join a lesson” section, enter a code that the teacher had previously given them. Students could access it via a mobile device, tablet, or computer. 

In addition to the theoretical content, the presentation was interspersed with different activities that served as a review of what had been explained previously. Questionnaires, free text tasks, “looking for pairs”, “time to climb”, and “fill in the blanks” were carried out. The same contents were given to the control group using the traditional method with a PowerPoint presentation.

At the end of the presentation, both groups were asked to complete a multiple-choice questionnaire through the virtual campus platform. There were two questionnaires with the same questions but with restrictions for students to answer within their assigned group, and both groups had 10 min to complete the 10-item questionnaire (Appendix A). There were multiple-choice, true-false, gap-fill, and matching questions. The students had only one attempt to complete the questionnaire.

### 2.2. Instruments and Outcomes

To assess students’ satisfaction with the applied learning method, the “Questionnaire of student’s satisfaction” (Cuestionario de Satisfacción del Discente in Spanish) (CSD) (Appendix B) was used as part of the evaluation tool ‘eValúa’, designed by the Continuous Professional Development Project of the Andalusian Agency for Health Quality (Spain) [33]. The CSD is presented as a reliable and valid tool to measure satisfaction with continuing health education. It includes 23 items in 5 dimensions (usefulness, methodology, organization and resources, teaching skills, and overall assessment). The scale of this survey is 0 representing the “lowest degree of satisfaction or strongly disagree”, and 10, the “highest degree of satisfaction or strongly agree”. The CSD was found to be highly reliable, with an overall Cronbach’s α of 0.979 [33]. This questionnaire was provided to students through a link located on the virtual campus of the subject, using the tool “Google Forms”. The data were downloaded onto an Excel sheet and analyzed.

A 13-question satisfaction survey (Appendix C) with more open-ended answers was also used to complement the previous questionnaire. The questions dealt with the changes detected with respect to the traditional method, the effectiveness of the tool, the interest in the methodology development, their opinion about involving more subjects, and the quantification of the time invested in relation to the level of knowledge acquired.

Knowledge was assessed using a post-test at the end of the theory class. A pre-test was intentionally avoided to prevent alerting participants to the material being tested. Two professors involved in the project developed the questions designed to assess the application of knowledge. Demographic information was ascertained using a baseline questionnaire. The satisfaction interview and CSD were conducted by the experimental group so that they could compare it with the traditional method.

### 2.3. Sample Size Calculation

Taking into account that the subject is followed each year by about 60 students, an alpha error equal to 5% and a power of 15% were assumed, obtaining a minimum sample size of 20 subjects per group [34].

### 2.4. Statistical Analysis

Knowledge scores (percent correct) and time spent (measured through the questionnaire tool on the online campus) were compared between the two groups using the Parametric model, two-factor Anova. A sensitivity analysis was conducted using the median of the preference scale for missing data. The responses to other questionnaires were analyzed similarly. All participants were analyzed in the groups to which they were assigned, and all data available for each participant were included. The significance level used was 0.05. All analyses were performed using IBM SPSS Statistics version 19 (IBM Inc., Armonk, NY, USA).

This study was carried out with the authorization of the Ethics Committee of the University of Cadiz (Spain), specifically the CEENB-GMOs Section (Ethics Committee for Non-Biomedical Experimentation and Evaluation of Experimentation with Genetically Modified Organisms) of the Bioethics Committee of the University of Cadiz (Ref. 001/2022).

All information related to the study was confidential, with only the researchers having access to the data, which guaranteed its treatment with the security measures established in compliance with Organic Law 3/2018 of 5 December that regulates the rights of Personal Data Protection and guarantees digital rights. All participants gave consent to use their data, and the data from consenting students were deidentified before analysis to protect student privacy.

## 3. Results

The satisfaction survey was completed by 26 students with an average age of 22 years, 57.1% of whom were male. A total of 83.33% thought that there were positive changes in terms of their dynamism and interactivity with this type of presentation compared to the traditional method, and 97.62% considered it effective in terms of attention gain and the use of this type of methodology had the possibility of increasing interest in other subjects. In terms of the perception of the students related to the level of interest that this teaching method provoked in them, they found it interesting (42.86%) or very interesting (54.76%). 

Approximately 88% thought that the time invested was adequate in relation to the level of knowledge acquired, and 97.62% considered that what they had learned with this type of practice had helped them to understand the contents of the subject.

The most repeated advantages described by the students were that it was dynamic, enjoyable, and interesting, that it increased attention and avoided boredom, it was interactive, and, therefore, entertaining and fun. It was innovative and helped to maintain concentration.

As far as the drawbacks are concerned, they were that you had to be very attentive, that some videos became monotonous (here they were assessing the content of the subject, not the teaching method), small technical problems with the internet, little time to answer the questions and that it could slow down the dynamics of the class.

Finally, as suggestions, they stated that the accessibility of the application and the pages used should be improved and that the time taken to answer the questions should be adjusted.

The CSD was answered by 23 students. In Figure 2, the answers obtained by the students to the questions referring to the “Utility” module can be observed. In all 3 items, the results are higher than 7. This is supported by the average of the first question, “The expectations I have fulfilled”, which provides a value of 8.8; the second question, “The contents developed have been useful”, has an average of 9.1, and the last question obtains an average of 9.06. All these data indicate that the surveyed students consider the application of Nearpod useful. 

Regarding the second dimension of the questionnaire, “Organisation and resources”, as in the previous case, most of the answers provide values higher than 7. In this case, the questions refer to “The didactic resources have been adapted to the optimal development of the activity”, which has had an average value of 9.09, and “The duration of the activity has been adequate to acquire the objective”, which has an average value of 8.74. Therefore, according to the answers provided by the students, both the resources and the time spent were adequate for the Nearpod activity.

The “Methodology” dimension includes the items “the teaching methods used by the teachers have been adequate for the optimal development of the activity, and “the evaluation system used allowed me to know my level of competence after the development of the activity”. The average data obtained in these two questions exceeds an average of 8 points, 8.8 and 9.06, respectively. Therefore, the students consider that this type of didactic presentation is suitable to be used in their learning and assessment. The opinions provided by the students indicate that the interest is mainly due to the fact that they can continue working at home and not only in the practical classes where they had to follow the teacher’s instructions. They can use Nearpod to continue studying and training at home, so their understanding of the techniques explained in the class is more appropriate.

Figure 3 shows students’ overall satisfaction and whether they would recommend the techniques used in class to other professionals. An average value of more than 9 points was obtained in both cases, which is considered very high. This indicates that students are very satisfied with the use of the application.

In relation to the activity reports, except for the questionnaire, which was answered correctly by 68% of the students, the other activities were answered correctly by more than 90%, which showed that they had understood what had been explained in the previous slides. Table 1 shows the descriptive statistics of the CSD.

As for descriptive ratings by gender, it does not have a great influence; the mean of the responses of men and women vary little (Table 2). The same happens for the intervention or control group (Table 3).

Regarding the relationship between the use of Nearpod and students’ academic performance, the analysis carried out in Table 4 (intersubject effects test) tries to find out if there are differences in the ratings of the two groups also using other covariates that modulate the process such as gender and time.

The consequence of the analysis is that:(1)sex does not influence at all, having a significance of 0.710 > 0.05, which is the confidence level (95%);(2)the group or treatment does not have any influence, as its significance is 0.075 > 0.05;(3)the response time does have an influence, as its significance is 0.022 < 0.05.

## 4. Discussion

The aim of this study was to determine students’ satisfaction with the interactive presentation tool Nearpod and to evaluate its influence on students’ performance in studying musculoskeletal disorders.

From the quantitative data collected in the satisfaction surveys, the general results showed encouraging comments from students regarding its use in the classroom to improve their learning experience.

To date, the most studied aspect of this type of presentation with Nearpod has been the degree of students’ satisfaction with its use, always with positive results [14,28,29,30,35] in agreement with our findings in Health Science students.

We agree with Shehata et al., whose project included a focus discussion group in which most of the comments were positive and few were neutral or negative, on recommending its use in other subjects [29]. In addition to students’ satisfaction, other findings were enrichment of the learning experience and improved interest [29], increased interaction between teachers and students [14], reduced distraction [36], increased student performance [31,37], and increased motivation [38]. This last element is crucial to support learning. Motivation is closely related to emotions because it reflects the extent to which an organism is prepared to act physically and mentally in a focused way. Thus, it can be argued that emotional systems create motivation that is conducive to learning [39]. Therefore, it is essential to incorporate new educational practices that respond to a new paradigm in which knowledge and emotion are founded in the classroom [40].

Concerning the teachers’ opinions about the experience, they commented that using it for the questionnaires was effective, as they saved on resources such as paper and time [29]. They also observed that with the discussion within the classroom that could be achieved with Nearpod, better learning was achieved, with higher involvement of less motivated students, as the real-time assessment caused them to feel more pressure due to the influence of the visualization of the results by their peers during the feedback process [14].

As regards students’ performance, to our knowledge, this is the first study in physiotherapy education to assess the academic performance of students through the comparison of a control group and an experimental group. Much of the existing research on this tool is limited to self-reported results related to satisfaction, and there is little work on objective measures such as students’ academic performance [31]. However, there is a study of physiotherapy students in which adding the Nearpod technology into the flipped classroom improved students’ performance on the final exam, compared to previous academic courses where the traditional model was used [41].

We have not found any studies analyzing the Nearpod tool in which response time has been taken into account as a variable in academic performance. Nevertheless, there are other ICTS that belong to the Student Response Systems (SRSs) or Learner Response Systems (LRSs), such as Kahoot, that reward the shortest response time [42].

With reference to how this tool was used, in our study, it was used as a presentation of a theoretical class reinforced with different questionnaires and activities that provided formative feedback on learning, coinciding with Shehata’s study [29]. However, in other research, it was used for real-time assessment [14], working in team activities [28,30], or as a complement to the Flipped Learning methodology [38].

As for the number of students, in our case, there were 28 students in the experimental group, but it was not carried out by the whole class. Mattei et al. [41] commented that it could be managed effectively in classes of 60 students, so for later courses, it could be implemented in larger groups.

Furthermore, we should take into account the technical problems encountered in certain research, such as McClean’s, where some of the Pharmacy and Bioscience students expressed their concern about the connection problems to the institutional Wi-Fi network [30], as well as in Ríos–Zaruma’s [35]. Other inconveniences encountered were complications when starting the presentation and the freezing of some sessions on some of the devices used [41]. In our case, although it was non-presential, four of them (9.3%) had the screen freeze. It is clear that good wireless coverage is important in the use of this type of technology.

There are few studies carried out on healthcare students on the use of this powerful digital tool that could be useful for them because, with this tool, they have the possibility of watching the presentation in deferred mode and, thus, are able to assimilate the contents at their own pace. It would also be very useful for this type of student to watch videos of the practical exercises that they will later carry out in person [43,44].

On the other hand, as mentioned above, the level of knowledge about ergonomic risks suggests a preventive aspect for musculoskeletal injuries, justifying their identification for the improvement of occupational health and safety [45], and this type of learning would be recommended to address the prevention of musculoskeletal disorders in health care workers [46].

### Limitations and Recommendations for Future Research

There are certain limitations in our work, such as the absence of a pre-test, the size of the sample divided into two groups, and technical problems. Although, at least in our case, it was not a virtual classroom that allowed us to appreciate these technical difficulties with all the students in the face-to-face class.

Future research is necessary, taking into account these limitations, in order to implement these types of technological tools that increase interactivity and assessment in real-time, with a stable internet connection to prevent technical problems.

When using a particular teaching strategy, there must be evidence that it is the best option for the purposes we are pursuing. This is also the focus of attention of Health Science educators [47]. Therefore, further research on this type of presentation is needed to improve teaching–learning interactions.

It would be advisable to carry out new studies with different subjects in various academic courses and in different centers to evaluate if these variables could influence the results. It is necessary to examine how diverse designs of the quizzes could affect students’ learning in order to adapt the ICT presentations, such as Nearpod, to the preferences of current students.

Our purpose was to conduct a pilot study to obtain preliminary reports in an intervention study with the Nearpod tool. We also wanted to determine the degree of satisfaction with the use of tools with a sample of physiotherapy students to check that they found it comfortable and useful for future interventions. Therefore, we encourage researchers to perform randomized controlled trials using larger sample sizes. Finally, we also urge researchers to analyze the effectiveness of the application of Nearpod in students or patients to identify the key aspects that could have a greater impact on its use in health education.

## 5. Conclusions

After the analysis, we can confirm the high level of student satisfaction with the use of the Nearpod application, becoming a useful practice for physiotherapy students to play an active role in their training, increasing their motivation in the teaching–learning process. There is no difference in test scores, although there is a reduction in the response time to the knowledge assessment questionnaires. Although the above are preliminary reports from a pilot study and further studies need to be carried out with larger samples and in different subjects to obtain more extrapolatable results, it seems that Nearpod is a tool with great potential for teaching in the study of musculoskeletal disorders.

## Figures and Tables

**Figure 1 ijerph-20-00099-f001:**
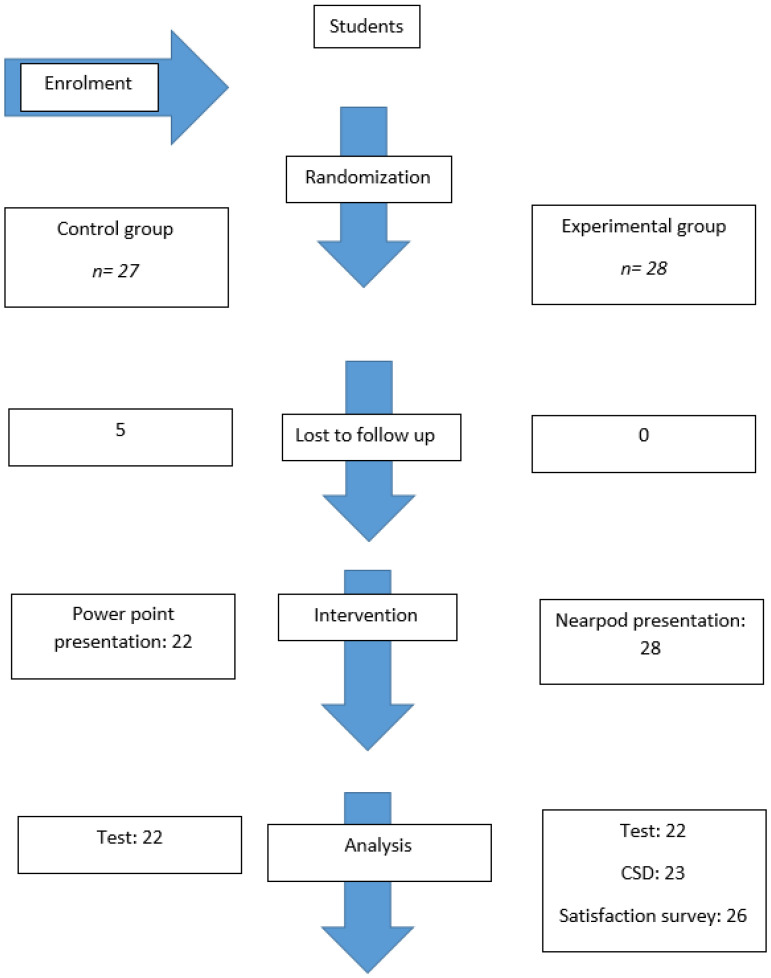
Flow diagram of student allocation.

**Figure 2 ijerph-20-00099-f002:**
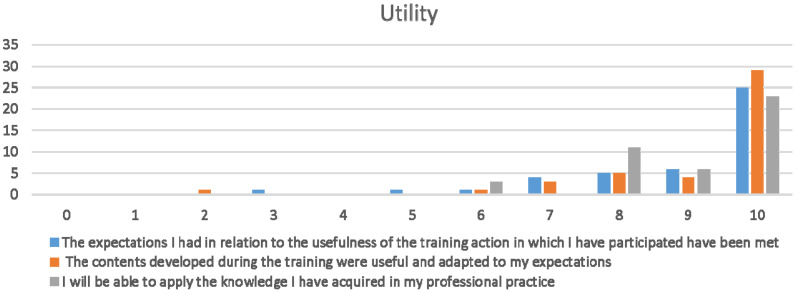
Utility dimension (CSD): x axe: strongly disagree or agree of satisfaction/y axe: number of responses.

**Figure 3 ijerph-20-00099-f003:**
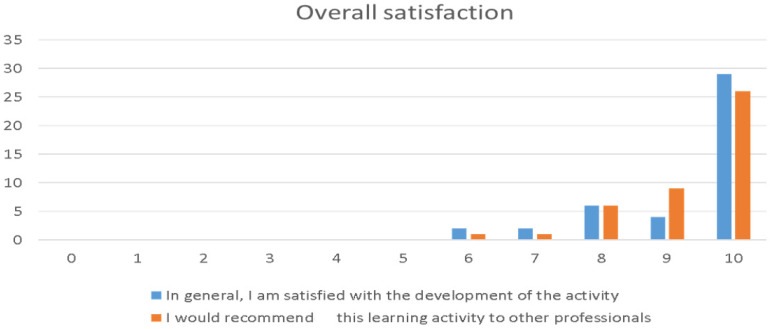
Overall satisfaction (CSD): x axe: strongly disagree or agree of satisfaction/y axe: number of responses.

**Table 1 ijerph-20-00099-t001:** CSD’s statistics.

Question	Valid	Missing	Media	Median	Mode	Standard Deviation
1	10	0	2.60	1.50	0	3.534
2	10	1	0.10	0.06	0	0.136
3	10	0	2.60	1.00	0	3.273
4	10	0	0.10	0.04	0	0.126
5	10	0	2.60	1.01	0	3.340
6	10	0	0.10	0.08	0	2.823
7	10	0	2.60	2.00	0	3.098
8	10	0	0.10	0.08	0	0.119
9	10	0	2.40	1.50	0	2.757
10	10	0	0.09	0.06	0	0.106
11	10	0	2.60	1.50	0	3.204
12	10	0	0.10	0.06	0	0.123
13	10	0	2.60	1.50	0	3.134
14	10	0	0.10	0.06	0	0.121
15	10	0	2.60	1.50	0	3.373
16	10	0	0.10	0.06	0	0.130
17	10	0	1.80	1.00	0	2.486
18	10	0	0.07	0.04	0	0.096
19	10	0	1.80	1.50	0	2.251
20	10	0	0.07	0.06	0	0.087
21	10	0	2.40	2.50	0	2.413
22	10	0	0.09	0.10	0	0.093
23	10	0	2.60	2.00	0	2.951

**Table 2 ijerph-20-00099-t002:** Descriptive scores by gender (marginal means).

Gender	Mean	Standard Error	95% Confidence Interval	
			Lower Limit	Upper Limit
Male	5.633 ^a^	0.321	4.988	6.279
Female	5.810 ^a^	0.349	5.108	6.512

^a^ The covariates appearing in the model are evaluated at the following values: time = 32,190.

**Table 3 ijerph-20-00099-t003:** Descriptive scores by learning group (marginal means).

Group	Mean	Standard Error	95% Confidence Interval	
			Lower Limit	Upper Limit
Itervention	6.199 ^a^	0.330	5.536	6.863
Control	5.244 ^a^	0.377	4.484	6.003

^a^ The covariates appearing in the model are evaluated at the following values: time = 32,190.

**Table 4 ijerph-20-00099-t004:** Tests for inter-subject effects.

Origin	Type III Sum of Squares	fd	Quadratic Mean	F	Sig
Corrected model	18.335	3	6.112	2.216	0.099
Intercept	52.997	1	52.997	19,213	0.000
Group	9.140	1	9.140	3313	0.075
Time	15.442	1	15.442	5598	0.022
Gender	0.386	1	0.386	0.140	0.710
Error	126.889	46	2.758		
Total	1810.908	50			
Total corrected	145.224	49			

## Data Availability

Not applicable.

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
