# Peer review of "Satisfaction Level and Performance of Physiotherapy Students in the Knowledge of Musculoskeletal Disorders through Nearpod: Preliminary Reports"

_ijerph, 2022, doi:10.3390/ijerph20010099_

Round 1

Reviewer 1 Report (Previous Reviewer 2)

The Authors made the required corrections.

Author Response

The Authors made the required corrections.

Reviewer 2 Report (New Reviewer)

Review Manuscript ID ijerph-2080093

First of all, I would like to congratulate the authors for their work.

Please, find below some comments and questions about the manuscript that have arisen after careful reading, and whose objective is to contribute to the improvement of the manuscript's readability.

 ·  Abstract. You wrote: “The participants were the students of the University of Cadiz”. You should specify that they were from the degree in Physiotherapy, and the year and course.  

 ·         What do the yellow highlighted paragraphs mean?

 ·         In Conclusions, they say that motivation improves. Did they measure it in some way in both groups, or was it only self-reported?

 ·         Indicate in reference 21, what type of document is. scientific article?, congress abstract or proceeding? I have recovered a related publication about Nearpod, in Physiotherapy, that you could review and consider in the references:https://files.eric.ed.gov/fulltext/EJ1143320.pdf. I comment on this because perhaps you should not be so categorical in your wording of lines 97-99 (to our knowledge, there are no articles that have analysed this type of tools in the Degree of Physiotherapy and neither in the study of musculoskeletal disorders).

 ·         You could include in the discussion section the role that motivation or emotion plays in increasing attention and learning during the classes. It is a critical issue to support your results partially

 ·         Lines 97-99 Please, explain the allocation process with envelopes. It is not clear to the reader in the present form.

 ·         Line 120. My question: was the Nearpod used only in one class session? how long did the session last?  Are the sessions for both groups given simultaneously o on different days? And in that case, was it the same teacher who took part in the sessions?

 Results

·         Line 195. The satisfaction survey was carried out by 26 students with an average age of 22 years, 57.1% of whom were male. 83.33% thought that there were important changes with this type of presentation compared to the traditional method and 97.62% considered it effective and with the possibility of increasing interest in other subjects with the use of this type of methodology.

Please, define important changes and effectiveness in this context.

-            Effective for? gain attention? improve learning? have fun? Please specify.

 ·           Line 200. In Figure 2 we can… be consistent with the wording of the rest of the manuscript using the third person (impersonal)

•     In table 3, the row of the control group would be missing, right? traditional group presentation

·      Line 283. Be uniform when citing: Health Science students – Health Students….use similar wording in all cases

 Thank you for your work. It has been a pleasure to read your study.

Congrats again, and good luck!  

Author Response

Title: Satisfaction level and performance of physiotherapy students in the knowledge of musculoskeletal disorders through Nearpod. Preliminary reports.”

Dear Editor and reviewers,

First of all, we would like to thank you for your comments and for allowing us to address the issues you raise to improve the manuscript’s quality. We appreciate your observations and the time devoted to the constructive criticism and feedback of our manuscript. Please find the answer to your comments below.

Response to reviewer 2 Coments.

 Point 1:

Abstract. You wrote: “The participants were the students of the University of Cadiz”. You should specify that they were from the degree in Physiotherapy, and the year and course. 

Response 1:

We have specified the required data.

Point 2:

 What do the yellow highlighted paragraphs mean?

Response 2:

It was a mistake. Sorry for the inconvenience.

Point 3: 

 In Conclusions, they say that motivation improves. Did they measure it in some way in both groups, or was it only self-reported?

Response 3:

 It was self-reported through satisfaction surveys where one of the items was to comment on the advantages and disadvantages of the teaching method. Among the advantages, the most repeated was that it was more motivating and dynamic.

Point 4:

 Indicate in reference 21, what type of document is. scientific article?, congress abstract or proceeding? I have recovered a related publication about Nearpod, in Physiotherapy, that you could review and consider in the references:https://files.eric.ed.gov/fulltext/EJ1143320.pdf. I comment on this because perhaps you should not be so categorical in your wording of lines 97-99 (to our knowledge, there are no articles that have analysed this type of tools in the Degree of Physiotherapy and neither in the study of musculoskeletal disorders).

Response 4:

  • The type of document was indeed not described. It was a proceeding. It has been corrected in the bibliography. Thank you for noticing.
  • We apologise for being so categorical. We have changed this sentence. Thank you very much for your input. This article was already among our references. We talked about it (page 11, line 401-403) in relation to the number of students using Nearpod: “Despite the fact that Mattei et al [42] commented that it could be managed effectively in classes of 60 students, so for later courses it could be implemented in larger groups”.

But thanks to the reviewer, we read it more closely and saw that they were indeed talking about physiotherapy studies. Thank you for your important contribution, we have added  another new paragraph on this article, recommended by the reviewer  in the discussion section, on page 11, lines 389-392. 

Point 5:

You could include in the discussion section the role that motivation or emotion plays in increasing attention and learning during the classes. It is a critical issue to support your results partially

Response 5:

We have introduced a paragraph in the discussion section on these important variables in the learning process (page 10, lines 370-376  ).Thank you very much for your recommendations.

Point 6.

 Lines 97-99 Please, explain the allocation process with envelopes. It is not clear to the reader in the present form.

Response 6.

The opaque envelopes were given to a university professor who was not involved in the project. He was in charge of shuffling the envelopes and handing them out randomly to the students.

This clarification has been added to the manuscript (page 3, lines 116-118). Thank you for improving this section.

Point 7.

 Line 120. My question: was the Nearpod used only in one class session? how long did the session last?  Are the sessions for both groups given simultaneously o on different days? And in that case, was it the same teacher who took part in the sessions?

Response 7.

Thank you for your input. We have added the following paragraph in the manuscript (page 3, lines 122-128) to make it clearer how the classes were organised:

“This project was carried out during two hours of theory. In one of the hours, the control group was with a subject teacher conducting a discussion on a topic unrelated to musculoskeletal disorders, while one of the project teachers was in another classroom with the experimental group using Nearpod.  The next hour, the control group went on to teach the musculoskeletal disorders class in the traditional way with the project teacher and the experimental group went to the second classroom to conduct the discussion with the other subject teacher”.

Point 8

Results

Line 195. The satisfaction survey was carried out by 26 students with an average age of 22 years, 57.1% of whom were male. 83.33% thought that there were important changes with this type of presentation compared to the traditional method and 97.62% considered it effective and with the possibility of increasing interest in other subjects with the use of this type of methodology. Please, define important changes and effectiveness in this context.

  • Effective for? gain attention? improve learning? have fun? Please specify.

Response 8:

- “positive changes in terms of their dynamism and interactivity”. It has also been specified in the text (page 6, lines 224-225). Thank you for your comments.

- Effectiveness was in terms of attracting attention. It has also been specified in the text on page 6, lines 225-226. Thank you for your input.

Point 9

 Line 200. In Figure 2 we can… be consistent with the wording of the rest of the manuscript using the third person (impersonal)

Response 9:

The impersonal form has been changed and used. Thank you for improving our work.

Point 10

   In table 3, the row of the control group would be missing, right? traditional group presentation

Response 10:

In table 3, in the control group row we refer to the group that followed the class of musculoskeletal disorders in the traditional way.

Group

Mean

Standar

error

 95% Confidence interval 

Lower limit

Upper limit   

intervention

6.199a

0.330                            

5.536

6.863

control

5.244a

0.377

4.484

6.003

Point 11

  • Line 283. Be uniform when citing: Health Science students – Health Students….use similar wording in all cases

Response 11:

Thank you for your suggestion. We have made appropriate changes to the manuscript to ensure uniformity.

   Please, do not hesitate to contact me, if you require further corrections and

information.

Thank you in advance

Round 2

Reviewer 2 Report (New Reviewer)

Thank you for responding to the comments and questions made. I consider that the modifications made are adequate and contribute to improving the readability of the manuscript.

I call on the authors to continue and expand this line of teaching innovation

This manuscript is a resubmission of an earlier submission. The following is a list of the peer review reports and author responses from that submission.

Round 1

Reviewer 1 Report

The authors have conducted a study focused on academia and the educational field. Although the study may present interesting results for students or teachers, the sample size and the general potential audience which may be interested in this study make the manuscript of low impact and scientific soundness. Furthermore, I think the manuscript does not fit with the topic of the Special Issue Physiotherapy and Exercise Rehabilitation for Patients with Musculoskeletal Disorders

1. The authors aimed to find out the degree of satisfaction of Physiotherapy Degree students with the use of this interactive tool and to assess the influence of using Nearpod in class on students’ performance while dealing with the topic of musculoskeletal disorders.
2.
Similar current research exists on this topic, thus the present study is not original or relevant to the field.

3. As aforementioned, there is a lack of novelty in the present study, and the small sample size makes the results hard to extrapolate.

4. The sample size is an important improvement to consider for future research. The design of the study must be presented following its characteristic guideline.

5. The characteristics of the present study make weak conclusions. The authors confirm their results, nevertheless, they should be shown, as they show at the end of their conclusion heading.

Reviewer 2 Report

In work entitled ‘Satisfaction level and performance of physiotherapy students in the knowledge of musculoskeletal disorders through Nearpod’, the authors described an important topic regarding the effectiveness of teaching and satisfaction using interactive tools such as Nearpod. The biggest limitation of the work is the small number of respondents, which significantly lowers the assessment of the quality of the research conducted.

The article requires a few changes.

TITLE

I suggest adding information on 'preliminary reports '.

INTRODUCTION

1. Lines 39-61 do not match the rest of the text. The authors discuss musculoskeletal disorders in the context of occupational diseases in physiotherapists. In the research part, there is no data on the physical examination of the musculoskeletal system in physiotherapy students in the context of possible future occupational diseases. Therefore, in my opinion, this part of the text should be changed. I suggest discussing interactive tools for learning their pros and cons in the education of physiotherapists.

MATERIAL AND METHOD

1. Research tools are correctly selected.

2. Material - too small study and experimental group. The research should be extended to a larger number of subjects.

RESULTS

1. Fig.2 and 3 have no description of the x and y axes.

DISCUSSION

Is not interesting, I suggest you improve it.